# Animal Virus Ecology and Evolution Are Shaped by the Virus Host-Body Infiltration and Colonization Pattern

**DOI:** 10.3390/pathogens8020072

**Published:** 2019-05-25

**Authors:** Jan Slingenbergh

**Affiliations:** Formerly Food and Agriculture Organization of the United Nations, 00153 Rome, Italy; slingenberghj@gmail.com

**Keywords:** virus ecology, virus evolution, livestock, virus transmission, generalists, specialists

## Abstract

The current classification of animal viruses is largely based on the virus molecular world. Less attention is given to why and how virus fitness results from the success of virus transmission. Virus transmission reflects the infection-shedding-transmission dynamics, and with it, the organ system involvement and other, macroscopic dimensions of the host environment. This study describes the transmission ecology of the world main livestock viruses, 36 in total, a mix of RNA, DNA and retroviruses. Following an iterative process, the viruses are virtually ranked in an outer- to inner-body fashion, by organ system, on ecological grounds. Also portrayed are the shifts in virus host tropism and virus genome. The synthesis of the findings reveals a predictive virus evolution framework, based on the outer- to inner-body changes in the interplay of host environment-transmission modes-organ system involvement-host cell infection cycle-virus genome. Outer-body viruses opportunistically respond to the variation in the external environment. For example, respiratory and enteric viruses tend to be associated with poultry and pig mass rearing. Ruminant and equine viruses tend to be more deep-rooted and host-specific, and also establish themselves in the vital inner-body systems. It is concluded that the framework may assist the study of new emerging viruses and pandemic risks.

## 1. Introduction

Animal viruses may be split into transmissible and persistent viruses. It has been proposed that transmissible viruses correlate with replication and virulence and that virus persistence instead permits a lower transmission rate [1]. This insight builds on earlier work suggesting that virus persistence may pave the way for virus-host symbiosis [2]. Viruses are considered essential agents within the roots and stems of the tree of life [3]. For example, RNA viruses in vertebrates tend to broadly follow the evolutionary history of their hosts that began in the ocean and extended for hundreds of millions of years [4]. The symbiotic virus-host relationships can take many forms, from antagonistic to mutualistic, and viruses, like other symbionts, lie on a continuum that can shift with environmental changes [5].

The present study seeks to take these insights to the next level. Starting point in the analysis is the link between virus propagation and transmission success. Virus transmission may be considered to present the backbone of virus ecology, determining the viruses selected for. Unfortunately, the current classification of animal viruses emphasizes the importance of virus genomic architecture and the host cell infection cycle [6]. Less attention is given to why and how virus fitness results from the virus transmission success. For example, an animal virus may become established in the upper respiratory tract and transmit via aerosols to the next host. An enteric virus features a fecal-oral cycle. A skin virus may transmit on the basis of touch. A virus colonizing the distal urogenital tract may transmit during sexual contact. Hence, an analysis of the virus transmission success requires a consideration of the overt clinical signs of infection, the gross pathology, the matching virus shedding profile, and of the ensuing modes of transmission.

The current analysis explores how the virus molecular world and the macroscopic dimensions of the host environment are intertwined, integral to one and the same virus transmission ecology. In vertebrate hosts the more vital organ systems are shielded off from external aggressors, small or large. It may be assumed that also the host immune defenses are structured to ensure that harmful viral pathogens remain confined to the outer-body environment, the epithelia. Epithelial viruses interface with the external environment and respond to the variation encountered here. Opposingly, infiltrative viruses establish in the inner-body environment and are expected to evolve towards a more intimate virus-host relationship. Given that the virus-host relationship changes with the position of the virus in the outer- to inner-body continuum the analysis focuses on the extent of virus host-body infiltration.

The virus organ system tropism is assumed to evolve in harmony with the virus cell tropism. This may be inferred from the dichotomy in the release of viruses from epithelial cells. Apart from direct cell-to-cell transmission [7], viruses may be released from the apical cell surface and so end-up in the outer-body environment. These viruses colonize mucosae and skin. In contrast, viruses released from the basolateral cell surface infiltrate underlying tissues. These viruses end-up in the lymph drainage, enter the blood circulation and so may infect any of the internal organs. Infiltration of and establishment in the inner-body environment may translate in additional, non-epithelial transmission modes. For example, virus establishment in the reproductive organs may translate in intrauterine or lactogenic transmission [8]. In birds, virus may be shed into yolk or albumen and so transmit vertically [9]. Virus circulation in the bloodstream may enable virus transmission via needles or arthropod vectors [10].

Taking an invasion ecology perspective, the host-body is viewed as a mosaic of organ systems. Viruses and organ systems are virtually portrayed in an outer- to inner-body fashion, based on the outer- to inner-body shifts in the virus infection-shedding-transmission characteristics which, in turn, result from the shift in virus organ system tropism. It is assumed that the nature of the virus-host interaction changes with the position of the virus in the outer- to inner-body continuum. The study describes the transmission ecology of the world main livestock viruses. The rationale for selecting the world main livestock viruses relates to the host damaging effects of these pathogens, the overt clinical signs and gross pathology, translating in prominent virus shedding and obvious virus transmission modes. Moreover, because of the major economic impact of these diseases, the causative viruses and the corresponding infection-transmission dynamics have been well studied.

Placed in a wider perspective the analysis builds on the growing perception that viruses deserve to be viewed as evolving living entities [11,12,13]. As biological replicators viruses require a propagation strategy in order to become transmitted to the next host [14]. For this, a virus may turn host damaging or instead evolve a friendly, persisting virus-host relationship [15]. 

## 2. Results

### 2.1. Virus Persistence Presents a Measure for the Extent of Virus Host-Body Infiltration

The analysis entailed an iterative process. As a first step, the one-to-three scores allocated to the 36 livestock viruses for the four ecological variables were examined in more detail. The scores are shown in Appendix A. The variables comprise the extent of virus host-body infiltration, the length of the infection period, the infection severity level, and the virus environmental survival rate. The one-to-three infiltration score reflects the organ systems involvement in infection-transmission and concerns, respectively, virus transmission based on the involvement of the epithelia, transmission involving epithelia and internal organs, and transmission involving just internal organs. The score for the length of the infection period reflects, respectively, acute, acute plus persistent, and persistent infections. Likewise, the score for the infection severity level concerns a case fatality of less than one, one to ten, and above ten percent. The score for the virus environmental survival rate refers to the number of days that the virus remains infective outside the host-body, ranging from up to three, three to ten, to over ten days. With one exception, the variables did not increase or decrease in value together. Just the association between the extent of virus host-body infiltration and the length of the infection period was found to be monotonic. Spearman correlation yielded an R = 0.58 and *p* < 0.0005. It was thus found that viruses infiltrating internal organs either cause persistent infections or a combination of acute and persistent infections. Conversely, persistent viruses either colonize internal organs or a combination of internal organs and epithelia. Hence, the length of the infection period appears to present a measure for the extent of virus host-body infiltration. 

### 2.2. A Coarse, A–D Infiltration Ranking of Virus Families

As a next step, the eleven virus families in the study were grouped and ranked A–D on the basis of the infiltration scores allocated to the individual family viruses, see Figure 1. The transmission of the viruses belonging to the *Orthomyxoviridae* and the *Paramyxoviridae* was found to strictly result from the involvement of the epithelia. The transmission of the viruses belonging to the *Coronaviridae*, the *Picornaviridae* and the *Poxviridae* was in part modulated also by the internal organ systems. The transmission of the viruses belonging to the *Arteriviridae*, the *Flaviviridae*, the *Herpesviridae*, plus also the single infectious bursal disease virus (IBDV), resulted from epithelial modes as well as internal organ systems involvement. Finally, the transmission of the single bluetongue virus (BTV) plus the viruses belonging to the *Retroviridae* family either reflected the involvement of epithelia plus internal organ systems or of just internal organ systems. A Spearman correlation of the A–D virus family specific infiltration ranking and the length of the infection period scores yielded an R = 0.71 and *p* = 0. The result indicates that the interrelationships among virus families may be defined in ecological terms and that the virus families may be neatly lined up in an outer- to inner-body fashion, virtually. 

### 2.3. The Outer- to Inner-Body Shift in Virus Organ System Tropism

Next, the organ system tropisms of the viruses belonging to each family were collectively fitted and with the naked eye aligned with the Figure 1 line-up of families. For this, the within-group, alphabetical family order was adjusted to secure an optimal visual match. As indicated in Figure 2, from outer- to inner-body the virus organ system appears to shift from the respiratory plus the alimentary tract to the skin, the distal urogenital tract or cloaca, the peripheral nerves and ganglia, the reproductive organs system, the lungs, to the immune plus the circulatory systems. Hence, there are indications that both viruses and organ systems may be lined up in an outer- to inner-body fashion, virtually. 

### 2.4. A New, One-to-Four Infiltration Score Applicable to Individual Viruses

Next, the A–D family ranking was converted into a one-to-four virus infiltration score applicable to individual viruses. Further, these scores are shown in Appendix A. To make the scoring compatible with the A–D family ranking, the one-to-four scores reflects, respectively, virus transmission strictly based on epithelial modes, primarily based on epithelial modes, involving epithelia and internal organ systems, and primarily involving internal organ systems. There are several differences with the A–D family ranking shown in Figure 1. Among the viruses of family group B, TGEV, AEV, FMDV, and LSDV received a score of three for transmitting on the basis of the involvement of both epithelia and internal organs. PEV1 and SVDV of group B were considered primarily epithelial and so received a two score. Both these viruses are persistently shed in feces, including in the absence of clinical signs, indicating a systemic infection component. The herpesviruses of group C were split into two. BHV-1, DEV, EHV-3, and GaHV1 were considered primarily epithelial while EHV-1, GaHV-2, and SHV-1 were considered to involve epithelia and internal organs. Among the D group viruses ALV was considered to involve epithelia and internal organs, unlike CAEV, JSRV, and MVV, for which the involvement of the epithelia did not appear to contribute to the overall virus transmission success. The latter viruses were allocated a score of four.

When the new, one-to-four virus infiltration scores were matched to the scores for the length of the infection period Spearman R became 0.73, and *p* = 0. When the somewhat atypical, vector borne bluetongue virus was removed from the correlation, R remained 0.73 for the one-to-four scoring, became 0.74 for the A–D virus family specific ranking, and 0.62, with *p* = 8 × 10^−5^, for the one-to-three infiltration ranking. 

### 2.5. The Outer- to Inner-Body Line-Up of 36 Viruses 

#### 2.5.1. Shifts in the Virus Organ System Tropism and Corresponding Transmission Ecology

Next, all of the above findings were considered in conjunction with the literature data on the transmission ecology collated for each of the 36 viruses in Appendix A. Pieced together on this basis was an outer- to inner-body line-up of viruses by organ system or combination of organ systems, guided by the one-to-four virus infiltration score, the corresponding virus organ system tropism, the matching virus transmission modes, length of the infection and shedding periods, infection severity level, and virus environmental survival rate, see Figure 3 and, also, Appendix A. 

#### 2.5.2. Outer-Body Shifts

For the epithelial, outer-body viruses it turned out that the length of the infection and shedding periods, as well as the virus environmental survival rate generally increased from respiratory tract to alimentary tract to skin. The respiratory viruses transmitted on the basis of aerosols, direct contact or fomites. Alimentary tract viruses were found to transmit on the basis of a fecal-oral cycle, through direct contact, contamination of feed and water, or involving fomites, persons and vehicles. Viruses infecting both respiratory and alimentary tract featured a mix of these transmission modes. Mostly, these viruses caused rather severe infections. Among the skin viruses, the more infiltrative viruses affecting all layers of the skin caused slowly healing lesions. The transmission of these deep-rooted skin viruses was found to rely on abrasion or biting flies rather than on direct touch or on indirect contact, more typical for superficial skin lesions. Some of the epithelial viruses are shed in feces over a prolonged time period, also in the absence of clinical signs, and these infections were considered to feature a systemic component. Next, the epithelial herpesviruses establishing latently in peripheral nerves and ganglia were found to cause a recurrence or persistence of the mucosal and/or skin infection, including of the distal urogenital tract and external genitalia. 

#### 2.5.3. Inner-Body Shifts

Virus infiltration of the inner-body environment frequently implicated the genital tract or reproductive system in general. This was found to be the case for the RNA, the DNA and for the retroviruses in the study. Virus establishment in the reproductive system translated in seminal transmission, haphazard abortion, late term abortion, stillbirth, birth of infected, yet apparently healthy offspring or, also, lactogenic transmission. The vertical transmission modes were common among the utmost deep-rooted viruses, the viruses infiltrating also the immune and circulatory systems. Some of the utmost infiltrative viruses featured an absence of epithelial transmission modes and were environmentally labile. Virus infiltration of the immune system associated with immune-suppression, severe infections, neoplasia, or instead with in-apparent, persistent infection. Virus infiltration of immune and circulatory systems associated with iatrogenic transmission modes. Virus circulation in the bloodstream facilitated arthropod borne transmission. As indicated in Figure 3, the transmission of the bluetongue virus, the sole arbovirus in the study, was considered somewhat atypical because the virus usually causes a transient infection in the ruminant host while in midges remains infective for life. Hence, the involvement of the biological vector complicates a direct comparison with the transmission ecology of the remaining 35 viruses.

#### 2.5.4. Outer-Versus Inner-Body

The finding that virus environmental survival in the outer-body environment increased from respiratory tract to alimentary tract to skin and decreased with the shift from the outer- to the inner-body environment prompted a re-examination of the relationship between the extent of virus host-body infiltration and virus environmental survival. Virus infiltration scores two-to-four, running from primarily epithelial transmission, to transmission involving also internal organ systems, to transmission primarily involving internal organ systems, were matched to the one-to-three virus environmental survival rate scores, yielding an R = −0.59 and *p* = <0.0005. The indication that at least in broad terms the extent of virus inner-body infiltration correlated with a loss of virus robustness was applied in the virus ranking, along with the other factors. 

### 2.6. The Outer- to Inner-Body Shifts in Virus Host Tropism and Virus Genome

Furthermore, the outer- to inner-body shifts in virus host tropism and virus genome were examined. Underlined in Figure 3 are ruminant and equine viruses, contrasted to the remaining, poultry and pig viruses. Excluded from the host tropism correlations was the multiple-host FMDV. The remaining 35 viruses all formed part of either of the two virus host groupings. It was found that the RNA, DNA and retroviruses broadly line up in an outer- to inner-body fashion. Virus host tropism and the virus genome line-up were matched. Additional correlations concerned the virus host tropism and the four virus ecological variable scores, as well as the virus genome type and the four virus ecological variable scores.

The extent of the host-body infiltration was found to increase from RNA to DNA to retrovirus, with R = 0.48 and *p* < 0.005. From RNA to DNA to retrovirus the ruminant and equine viruses gained in prominence, with R = 0.40 and *p* < 0.05, and the infection severity level decreased, with R = −0.45 and *p* < 0.01. Moreover, the ruminant and equine viruses were found to cause less severe infections than the poultry and pig viruses, with Spearman R = −0.55 and *p* < 0.001. Hence, from outer- to inner-body, the virus genome type and host tropism appear to shift in concert, along with the infection-transmission dynamics. 

## 3. Discussion

### 3.1. A Predictive Framework for Animal Virus Evolution

The synthesis of the findings is presented in Figure 4. The host environment frames the virus transmission modes and, with it, explains the organ system involvement, the specifics of the host-cell infection cycle, and the virus genome. Vice versa, the virus life history explains the virus genomics, the host-cell infection cycle and, with it, the macroscopic level virus-host interactions and the host population ecology. The interplay of the host environment-transmission modes-organ system involvement-host cell infection cycle-virus genome changes from outer- to inner-body, resulting in two opposite virus evolution pathways, respectively for generalist and specialist type viruses.

Implied by Figure 4 is that the crowding conditions observed in poultry and in pig husbandry tend to attract horizontally transmitting respiratory and enteric viruses. The pathogenicity level of the viruses evolves to match the dynamics in host abundance and contact rate. At the molecular level, these RNA viruses become released from the apical surface of epithelial cells directly into the outer-body environment. Thus, proliferative virus replication, generalized infection of respiratory plus enteric mucosae, profuse virus shedding, and swift onward transmission all go hand-in-hand. 

A diametrically opposite scenario is given by relatively stable host environments observed in ruminant and equine husbandry, with parent stock and their young grazing together in the open, not unlike wild herbivore ecologies. The viruses attracted and selected for establish in the vital inner-body systems and transmit vertically, via needles or via bloodsucking arthropods. At the molecular level, virus establishment in the vital body systems is matched by low replication rates and minor or slowly evolving host damage. The utmost infiltrative viruses in the study are the retroviruses. In addition, some of other RNA viruses are deep-rooted.

The DNA viruses in the study take an intermediary position.

### 3.2. Opportunistic Viruses and Outer-Body Habitats 

It has been established that epithelial viruses are highly evolvable, more so than inner-body viruses [16]. Epithelial viruses are responsive to the dynamics in the environment external to the host-body. This may be illustrated on the basis of the genetically related virus pairs in the study. For example, the influenza virus circulating in horses (EIV) generates a transient, dry cough supporting swift virus transmission via aerosols [17]. In pigs, the virus (SIV) causes coughing and sneezing, resulting from significant mucus production [18]. The virus transmits on the basis of close direct contact, in line with the social behavior and body size of pigs. 

The rinderpest virus (RPV) in cattle and buffaloes primarily colonizes the alimentary tract and transmits on the basis of direct muzzle-to-muzzle contact [19]. In small ruminants, the identical peste des petits ruminants virus affects also the respiratory tract and transmits also via aerosols. Likewise, the lumpy skin disease virus (LSDV) in cattle causes persistent, deep, necrotic skin plugs and transmits via biting insects, mechanically. In sheep and goats, the virus (SGPV) causes transient lesions [20].

### 3.3. Specialist Viruses and Inner-Body Habitats

The caprine arthritis-encephalitis virus (CAEV) and the Maedi-Visna virus (MVV) present an example of closely related lentiviruses establishing in the inner-body organ systems of sheep and goats. The viruses display overlap in host tropism and both transmit mainly vertically via colostrum and milk. The difference between the two viruses mainly concerns the differential inner-body virus organ system tropism.

Projected on a long evolutionary timescale, inner-body viruses tend to become locked in within the host body. This internalization may turn progressive when the epithelial transmission modes are being replaced by internal organ system-based modes. Virus establishment in the reproductive system translates in vertical transmission, in turn enhancing virus-host co-evolution [21]. Virus infiltration of also immune and circulatory systems may yield in-apparent, persistent infections, indicating low levels of pathogenicity and/or enhanced host tolerance. The division between virus and host may become blurred and given enough time the two may become one [22].

### 3.4. Predicting Species Jumps

The nature of species jumps differs between generalist and specialist type viruses. For example, an opportunistic, epithelial virus of wildlife origin is likely to be found circulating in livestock before becoming first detected in humans as host. This has been the case for influenza [23], Henipah [24] and MERS corona viruses [25]. Further, the SARS corona virus infected civet cats raised as food animals before appearing in humans as host [26]. In contrast, more infiltrative viruses establish in the vital inner-body systems. Specialist viruses circulating in the bloodstream of non-human primates may directly jump to humans as host, as a result of complex ecological, socio-economic, demographic and other drivers. Examples comprise HIV-aids [27], Chikungunya [28], and Zika viruses [29]. Hence, knowing how species jumps differ for the different host ecologies may assist the study of pandemic risks.

## 4. Materials and Methods 

### 4.1. Selection of the World Main Livestock Viruses

A subtotal of 23 livestock viruses of global animal health significance was extracted from the OIE-Listed diseases, infections and infestations in force in 2019 [30]. Livestock infections and diseases resulting from virus spill-over from wildlife were excluded from the analysis. The common livestock hosts, described in the colloquial OIE terminology, comprise horses, donkeys, cattle, buffaloes, sheep, goats, swine, chicken, turkeys, ducks, and geese. The selected viruses, again in OIE terminology, comprise, in alphabetical order:-infection with Aujeszky disease virus;-avian infectious bronchitis virus;-avian infectious laryngotracheitis virus;-infection with avian influenza virus;-infection with bluetongue virus;-bovine viral diarrhea virus;-caprine arthritis/encephalitis virus;-infection with classical swine fever virus;-infection with equid herpesvirus-1;-infection with equine arteritis virus;-equine infectious anemia virus;-equine influenza virus;-enzootic bovine leucosis virus;-infection with foot and mouth disease virus;-infectious bovine rhinotracheitis/infectious pustular vulvovaginitis virus;-infectious bursal disease or Gumboro disease virus;-infection with lumpy skin disease virus;-infection with Newcastle disease virus;-infection with peste des petits ruminants virus;-infection with porcine reproductive and respiratory syndrome virus;-infection with rinderpest virus;-sheep pox and goat pox virus;-transmissible gastroenteritis virus.

Added to the above were ten globally important livestock viruses drawn from the last, 1995, edition of the FAO-OIE-WHO Animal Health Yearbook [31]: -avian encephalitis virus;-avian leucosis virus;-contagious pustular dermatitis virus;-duck virus enteritis or duck plague virus;-enterovirus encephalomyelitis or Teschen disease virus;-fowlpox virus;-Maedi-Visna virus;-Marek’s disease virus;-pulmonary adenomatosis or jaagsiekte virus;-swine vesicular disease virus.

Furthermore, three viruses of international veterinary relevance were added:-equine herpesvirus-3;-porcine epidemic diarrhea virus;-swine influenza virus.

The total of 36 livestock viruses belong to eleven different families and form a mix of RNA (N = 19), DNA (N = 11), and retroviruses (N = 6).

### 4.2. Details on the Transmission Ecology and Virus Ecological Variables

Shown in Appendix A for each of the 36 viruses are the virus family, virus genomic architecture, virus name in full, abbreviated, and the common names given to the infection or disease. Also presented is a brief summary on the transmission ecology for each virus, with references to the primary livestock host, the virus organ system tropism, the length of the infection and shedding period, the infection severity level, the transmission modes, and the virus environmental survival rate. 

Presented in Appendix A are one-to-three scores allocated to the 36 viruses for four ecological variables. The variables comprise the extent of virus host-body infiltration, the length of the infection period, the infection severity level, and the virus environmental survival rate. The one-to-three infiltration score reflects the organ system involvement in infection-transmission and concerns, respectively, virus transmission based on the involvement of the epithelia, transmission involving epithelia and internal organs, and transmission involving just internal organs. Also shown is a one-to-four virus infiltration score, an outcome of the iterative analysis process and reflecting, respectively, virus transmission strictly based on epithelial modes, primarily based on epithelial modes, involvement of epithelia and internal organ systems, and of primarily internal organ systems. The score for the length of the infection period reflects, respectively, acute, acute plus persistent, and persistent infections. Likewise, the score for the infection severity level concerns a case fatality of less than one, one to ten, and above ten percent. The score for the virus environmental survival rate refers to the number of days that the virus remains infective outside the host body, ranging from up to three, three to ten, to over ten days. Also indicated in Appendix A is the virus host range as observed in both livestock and wildlife. 

Appendix A lists the literature sources on which Appendix A is based.

### 4.3. The Analysis 

The analysis concerned an iterative process. As a first step, the one-to-three scores allocated to the 36 viruses for the four ecological variables were examined in more detail and the monotonic associations subjected to Spearman correlation. Just the scores for the virus host-body infiltration and for the length of the infection period were found to increase in value together. 

Next, given the coarse match between virus infiltration and persistence, it was examined how this relationship played out at the virus family level. For this, the eleven virus families in the study were grouped and ranked A–D on the basis of the one-to-three infiltration scores allocated to the individual family viruses. Also, this A–D infiltration ranking was held against the length of the infection period scores.

Next, given the indication, from the above, that the virus families may be neatly lined up in an outer- to inner-body fashion, it was examined how the organ system tropisms of the family viruses aligned with it. For this, the organ system tropisms of the viruses belonging to each family were collectively fitted and with naked eye aligned with the A–D family groups. For this, the alphabetical family order within the family groups was abandoned in order to obtain an optimal visual match. The result confirms that also the organ systems may be lined up in an outer- to inner-body fashion, virtually.

Next, the A–D family ranking was converted into a one-to-four virus infiltration score applicable to individual viruses, as described in Section 4.3, and also these scores were matched to the length of the infection period scores.

Next, since the infiltration-persistence match for the individual viruses was found to be about as strong as for the virus families, the 36 viruses were individually lined up in an outer- to inner-body fashion, irrespective the family origin, strictly on ecological grounds. For this, all of the above obtained results were considered in conjunction with the literature data on the transmission ecology collated for each of the 36 viruses in Appendix A. Pieced together on this basis was an outer- to inner-body line-up of viruses by organ system or combination of organ systems, guided by the one-to-four virus infiltration score, the corresponding virus organ system tropism, the matching virus transmission modes, length of the infection and shedding periods, infection severity level, and virus environmental survival rate.

The finding that virus environmental survival in the outer-body environment increased from respiratory tract to alimentary tract to skin and decreased with the shift from the outer- to the inner-body environment prompted a re-examination of the relationship between the extent of virus host-body infiltration and virus environmental survival. Virus infiltration scores two-to-four, running from primarily epithelial transmission, to transmission involving also internal organ systems, to transmission primarily involving internal organ systems, were found to match with the one-to-three virus environmental survival rate scores. The indication that, at least in broad terms, the extent of virus inner-body infiltration correlated with a loss of virus robustness was applied in the virus ranking, along with the other factors. 

Next, furthermore examined were the outer- to inner-body shifts in virus host tropism and virus genome. For this, the ruminant plus equine viruses were contrasted to the poultry plus pig viruses. Excluded from the host tropism correlations was the multiple host FMDV. The remaining 35 viruses all formed part of either of the two host groupings. It was found that RNA, DNA and retroviruses broadly line up in an outer- to inner-body fashion. Virus host tropism and the virus genome line-up were matched. Additional correlations concerned virus host tropism and the four virus ecological variable scores, as well as virus genome type, and the four virus ecological variable scores. It was found that from outer- to inner-body, the virus genome type and host tropism appear to shift in concert, along with the infection-transmission dynamics.

The collective results above served the compilation of the predictive framework for animal virus evolution shown in Figure 4, Discussion section. 

### 4.4. Software

The online Rho calculator https://www.socscistatistics.com/tests/spearman/Default.aspx was used for the Spearman correlations. This software has been audited by established statistics packages.

## Figures and Tables

**Figure 1 pathogens-08-00072-f001:**
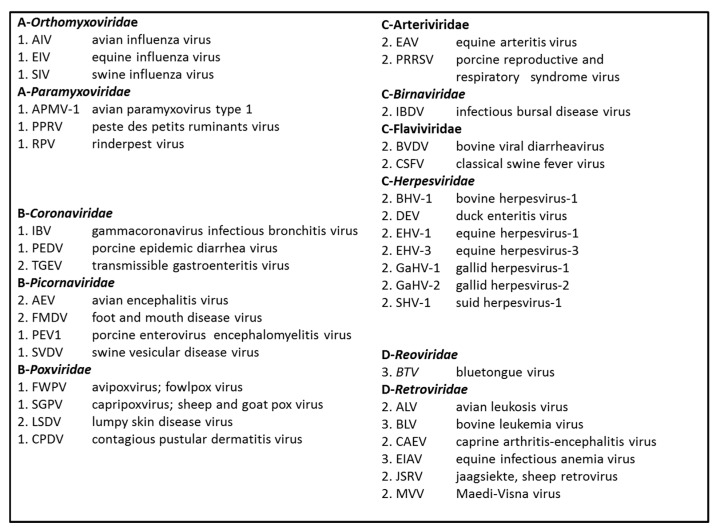
Virus family based host-body infiltration ranking. The eleven virus families in the study were grouped and ranked A–D on the basis of the host-body infiltration scores allocated to the individual family viruses. For each grouping, the virus families and also the family viruses are shown in alphabetical order.

**Figure 2 pathogens-08-00072-f002:**
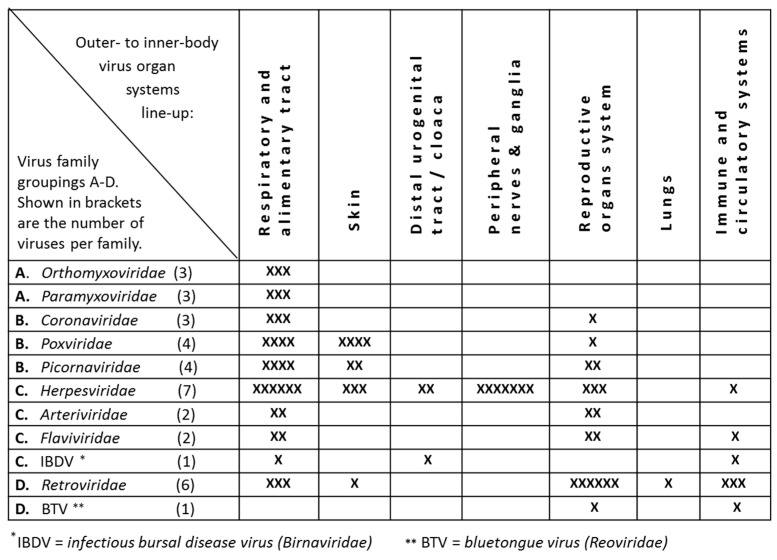
Organ systems virtually lined-up in an outer- to inner-body fashion. For this, the organ system tropisms of the viruses belonging to each family were collectively fitted and with the naked eye aligned with the Figure 1 A–D line-up of virus families.

**Figure 3 pathogens-08-00072-f003:**
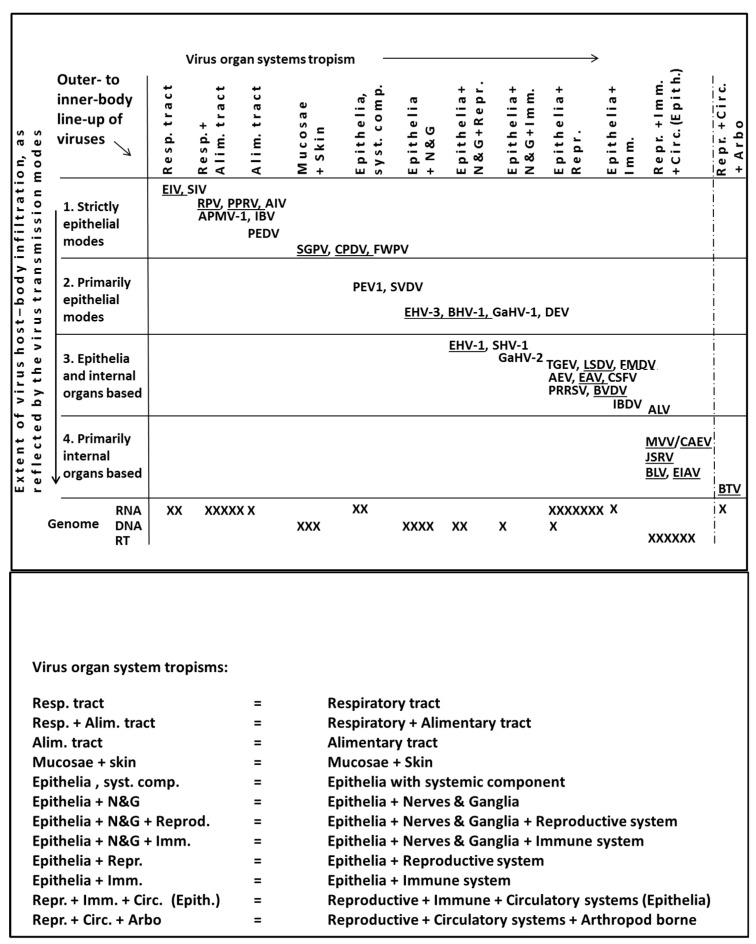
The 36 viruses in the study lined up in ecological terms. The first panel concerns an outer- to inner-body alignment of organ systems and viruses, plus also of virus genomes. Underscored are ruminant and equine viruses, so as to create a contrast with the remaining poultry and pig viruses, see text. Shown in the second panel are the organ system tropisms spelled out in full. See Figure 1 for the virus names in full.

**Figure 4 pathogens-08-00072-f004:**
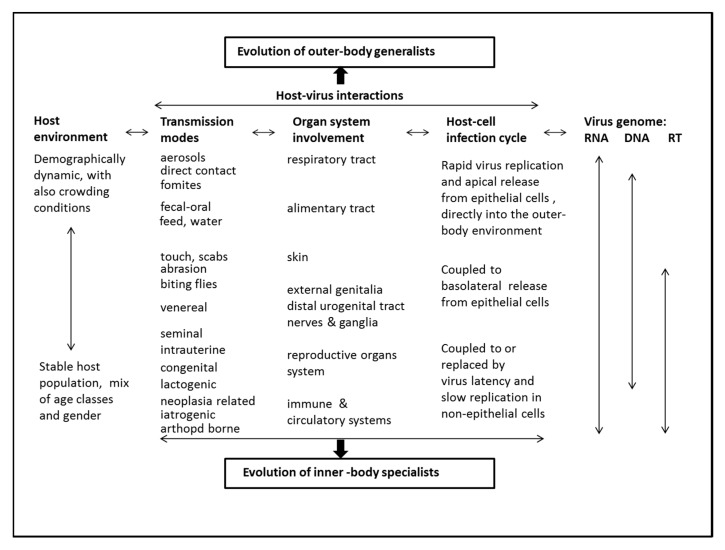
Framework for animal virus evolution. The framework is based on the outer- to inner-body shifts in the interplay of host environment-transmission modes-organ system involvement-host cell infection cycle-virus genome, resulting in two opposite virus evolution pathways.

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
