# Peer review of "Animal Virus Ecology and Evolution Are Shaped by the Virus Host-Body Infiltration and Colonization Pattern"

_pathogens, 2019, doi:10.3390/pathogens8020072_

Round 1

Reviewer 1 Report

This manuscript ‘animal virus ecology and evolution are shaped be the host body infiltration and colonization pattern” describes the host environment and their critical influence in spreading the virus. Is a well written easy to understand paper with the concept being clearly explained. Author describes the role of host inner and outer body interaction alter the infectivity and are well classified to different sub sections such as outer body shifts, outer vs inner and inner body shifts.

The manuscript needs more references in introduction and discussion part.

Line 44 need a reference to the lactogenic transmission.

In Discussion section add reference to the following lines.

 Line -247-248

Line 267-268 equine virus

Line 268-270 pigs

Line 271-272 RPV

Line 275 SGPV

Line 286-287 virus host evolution

Materials and method: Give more details on the methods of analysis.

 Line 385 Host body infiltration was found to be increased, any percentage increase or fold?

Reviewer 2 Report

- the scoring systems are not well explained, e.g. it is unclear what a score of 1 vs a score of 3 means in 2.1.

- The author does not explain why certain viruses get certain scores, this needs to refer back to all the literature the scoring is based on.

- It is not clear why the author chose the viruses in the paper, and not other livestock viruses

- The text is written like the virus makes informed decisions about transmission routes and tissues to infect, which is not correct.
